# Processing and Characterization of a New Quaternary Alloy Ti_10_Mo_8_Nb_6_Zr for Potential Biomedical Applications

**DOI:** 10.3390/ma15238636

**Published:** 2022-12-03

**Authors:** João Pedro Aquiles Carobolante, Adelvam Pereira Júnior, Celso Bortolini Junior, Kerolene Barboza da Silva, Roberta Maia Sabino, Ketul C. Popat, Ana Paula Rosifini Alves Claro

**Affiliations:** 1School of Engineering and Sciences, Guaratinguetá Campus, São Paulo State University (Unesp), Guaratinguetá 12516-410, Brazil; 2School of Advanced Materials Discovery, Colorado State University, Fort Collins, CO 80523, USA; 3Department of Mechanical Engineering, Colorado State University, Fort Collins, CO 80523, USA; 4School of Biomedical Engineering, Colorado State University, Fort Collins, CO 80523, USA

**Keywords:** titanium alloys, mechanical properties, Ti-Mo-Nb-Zr, biomaterial

## Abstract

The study of new metallic biomaterials for application in bone tissue repair has improved due to the increase in life expectancy and the aging of the world population. Titanium alloys are one of the main groups of biomaterials for these applications, and beta-type titanium alloys are more suitable for long-term bone implants. The objective of this work was to process and characterize a new Ti_10_Mo_8_Nb_6_Zr beta alloy. Alloy processing involves arc melting, heat treatment, and cold forging. The characterization techniques used in this study were X-ray fluorescence spectroscopy, X-ray diffraction, differential scanning calorimetry, optical microscopy, microhardness measurements, and pulse excitation technique. In vitro studies using adipose-derived stem cells (ADSC) were performed to evaluate the cytotoxicity and cell viability after 1, 4, and 7 days. The results showed that the main phase during the processing route was the beta phase. At the end of processing, the alloy showed beta phase, equiaxed grains with an average size of 228.7 µm, and low Young’s modulus (83 GPa). In vitro studies revealed non-cytotoxicity and superior cell viability compared to CP Ti. The addition of zirconium led to a decrease in the beta-transus temperature and Young’s modulus and improved the biocompatibility of the alloy. Therefore, the Ti_10_Mo_8_Nb_6_Zr alloy is a promising candidate for application in the biomedical field.

## 1. Introduction

Among the research areas within biomaterials is the development of new alloys [1]. An important application of metallic biomaterials is their use in medical devices to aid musculoskeletal disabilities, which are the main cause of life with disability [2]. These disabilities are more frequent in elderly people, and the global trend is for an increase in life expectancy and an aging population [3]. Therefore, the development of new metal alloys with more suitable properties remains an important area of study.

Titanium and its alloys are one of the most important groups of metallic biomaterials, mainly in hard tissue replacement [1,4]. Titanium is a corrosion-resistant metal with excellent biocompatibility, and its mechanical strength is comparable to that of cobalt alloys and stainless steel, but it has the advantage of being less dense and having a lower Young’s modulus [5]. When medical devices are implanted in the bone, it is convenient to use materials with Young’s modulus closer to that of the tissue (<30 GPa), which avoids the stress shielding effect that can lead to implant failure [6]. Studies have shown that beta-type titanium alloys with alloying elements that minimize martensitic phase formation can exhibit this characteristic. Consequently, complex systems have been designed and characterized [7,8].

The Ti-Mo-Nb-Zr system is a recent proposal, and little has been studied. However, prior systems allow us to understand the influence of these elements on the alloy’s behavior. Ti-Mo and Ti-Nb systems have been studied since the 1980s [9]. Molybdenum and niobium are beta-stabilizers and have been used to develop new alloys with low Young’s modulus and greater resistance to corrosion [10,11]. Molybdenum also induces increased flexural strength [12]. There is a controversy regarding the biocompatibility of molybdenum; however, many studies indicate that levels of toxicity and cell growth are suitable for biomedical applications [1,4,13]. There is the presence of martensitic phases in Ti-Mo alloys with a content of less than 10%wt of molybdenum. Thus, thermomechanical treatments can be used to increase fatigue strength and hardness [14]. The existence of martensitic phases allows some Ti-Mo and Ti-Nb alloys to have a shape memory effect and superelasticity [15,16].

Zirconium promotes the stabilization of the beta-phase in the presence of beta-stabilizers, despite being classified as a neutral alloy element [15,17,18]. Titanium alloys with high zirconium content are possible due to the similarities between the physicochemical properties of these elements [19]. In binary alloys, there is an increase in corrosion resistance, biocompatibility, and mechanical resistance (up to 40% higher) [20,21], a decrease in melting temperature, and an increase in hardness [22]. As niobium, zirconium increased the toughness of titanium alloys [23,24,25]. Some studies have indicated that zirconium suppresses the formation of the omega-phase [13,26]. Increased surface energy and improved cell development (osteoblasts) have also been reported when compared to CP Ti and Ti-Nb alloys [25].

Ti-Mo-Nb alloys have been evaluated as an option for biomedical applications [27,28,29,30,31,32]. The study of Ti_10_Mo_3_Nb, Ti_10_Mo_7_Nb, and Ti_10_Mo_10_Nb alloys indicated a microstructure with equiaxial grains, a beta-phase in the alpha-matrix, and an increase in the content of the beta-phase with the increase in the concentration of niobium in the composition. Regarding the mechanical properties, there was a decrease in hardness (394-444 HV) with the increase in the beta-phase content; however, the values are still higher than CP Ti. Young’s modulus (28.4–24.7 GPa) also decreased with increasing niobium content [27]. An increase in the corrosion resistance of the analyzed alloys was observed, despite the niobium contributing to the occurrence of the beta-phase, which is less resistant than the alpha-phase. It is proposed that the decrease in the number of vacancies in the passive layer has promoted an increase in corrosion resistance [28,29]. In vitro studies showed strong adhesion, good proliferation, and cell differentiation results like those for CP Ti. The Ti-Mo-Nb system also showed no cytotoxicity or inflammatory response [30,32].

Alloy systems with molybdenum, niobium, and zirconium have been successfully developed, mainly for orthopedic and dental applications. Ti-Nb-Cu [7], Ti-Zr-Nb [13,22,33,34,35], Ti_33.6_Nb_4_Sn [8], Ti_17_Nb_6_Ta [36], Ti-Mo-Nb [14,16,27,28,29,30,31,37,38], Ti_25_Nb_3_Zr_3_Mn_3_Sn [39], Ti-Mo-Zr-Fe, Ti-Mo-Nb-O, Ti-Nb-Zr, Ti-Nb-Zr-Ta, Ti-Zr-Mo-Mn [40], Ti-Mo-Zr [40,41] are examples of recent studies that show good results in terms of mechanical properties, biocompatibility, corrosion resistance, osseointegration, and Young’s modulus.

There are few reports concerning the Ti-Mo-Nb-Zr quaternary system. The alloys Ti_8_MoXNbXZr alloys (X = 2, 3, 4, 5, 6% at.) exhibited superelasticity and low Young’s modulus [42]. The increase in niobium content increased tensile strength, with better specific resistance for Ti_8_Mo_5_Nb_3_Zr alloy. In general, zirconium and niobium favored a decrease in Young’s modulus and an increase in toughness and deformation. Therefore, our aim in the present study was to evaluate the influence of zirconium addition on the properties of the ternary system Ti_10_Mo_8_Nb previously evaluated by our group [31,32,43]. Mechanical and microstructural characterization was performed for the quaternary alloy Ti_10_Mo_8_Nb_6_Zr, and its potential for biomedical applications was evaluated.

## 2. Materials and Methods

### 2.1. Alloy Processing

Ti_10_Mo_8_Nb_6_Zr alloy was obtained from commercially pure titanium (grade 2), molybdenum, niobium, and zirconium. The metals were chemically etched in a solution composed of HNO_3_ and HF (3:1). The ingots were melted in a voltage arc furnace with a tungsten electrode, in a copper crucible cooled by a water stream, and in an inert atmosphere (Ar). Each ingot is remelted at least 10 times, followed by heat treatment in a tubular resistive furnace under a vacuum. The homogenization heat treatment was carried out at 1000 °C for 24 h, with a heating rate of 16 °C/min and non-forced cooling, inside the furnace. Then, annealing treatment was carried out at 950 °C for 2 h and quenched in water at room temperature. A heating rate of 15°C/min was used. In the next step of the processing route, the ingot was cold-forged until a cylinder with a diameter of about 10 mm was obtained. Finally, new quenching was carried out following the method previously described.

### 2.2. Alloy Characterization

The chemical composition of the alloy at the end of processing was verified by X-ray fluorescence spectroscopy (Axios MAX, PANalytical) in a semi-quantitative analysis without standards, with the determination of chemical elements from fluorine to uranium. The sample was prepared by setting it in boric acid.

XRD, optical microscopy, and microhardness measurements were performed after each step of the alloy processing route: as-cast, after homogenization heat treatment, after quenching treatment, after forging, and at the end of the processing.

X-ray diffractometry data were obtained from an Advange D8 (42 kV/120 mA) diffractometer, Bucker, with Cu-Kα radiation and Bragg–Brentano geometry. The 2θ measurement range was 20° to 100°, with a 0.02° step, 0.35 s counting time, 0.6 mm slip opening, 10 rpm sample rotation, 25 mA current, and voltage of 40 kV. The data was refined using the Rietveld method, using Topas software (Bucker). The phases were identified using the Inorganic Crystal Structure Database (ICSD) and with the Crystallographica Search-Match software (version 2.1.1.1, from Oxford Cryosystems).

Optical microscopy was performed using an Epiphot 200 microscope (Nikon, Tokyo, Japan). The samples were polished with silicon carbide sandpaper (up to #2000) and with colloidal silica solution. Dripping of the oxalic acid solution (3% *w*/*v*) was used to assist with the polishing. The chemical etching to reveal the microstructure was carried out by immersion of the samples in a solution composed of H_2_O, HNO_3_, and HF (85:10:5). ImageJ 1.52s software was used in the analysis of the micrographs.

A sample of 35.8 mg of Ti_10_Mo_8_Nb_6_Zr alloy at the end of the processing route was used to evaluate thermal behavior by differential scanning calorimetry (DSC) (STA 409 model, NETZSCH). Al_3_O_2_ crucibles were used in the analysis. Two heating-cooling cycles were performed. The sample was heated to 1000 °C and cooled to 100 °C at a rate of 10 °C/min. There was a flow of argon gas of 100 mL/min inside the chamber during the analysis.

The microhardness measurements were made on a microdurometer (6030 model, OMNIMET BUEHLER) with a Vickers diamond penetrator in the shape of a square-based pyramid with a 136° angle between planes, using a load of 50 g (490.5 μN) for 15 s. 20 measurements were performed per sample.

Young’s modulus was obtained by impulse excitation technique (Sonelastic, ATCP Engenharia Física, Ribeirão Preto, Brazil), following the standard ASTM E1876-15. NA AS-BS support was used to perform the measurements. Four samples of the Ti_10_Mo_8_Nb_6_Zr alloy at the end of the processing were used. Young’s modulus was obtained by measurements performed by both the flexural and longitudinal methods. Ten measurements were performed per sample in each of the methods used.

### 2.3. In Vitro Studies

In vitro studies were performed on CP Ti, Ti_10_Mo_8_Nb, and Ti_10_Mo_8_Nb_6_Zr alloys. CP Ti was used as a control due to its remarkable biocompatibility [44]. The cytotoxicity of adipose-derived stem cells (ADSC) was verified after 1 day of cell culture, and ADSC viability was investigated after 4 and 7 days of cell culture. Four discs (3 mm thick and 10 mm diameter) from each alloy were used to analyze cytotoxicity, and five discs were used for cell viability studies.

#### 2.3.1. Cell Culture

ADSCs were isolated from adipose tissue by Prof. Cox-York’s lab at the Department of Food Science and Human Nutrition at Colorado State University. The protocol for ADSCs isolation from healthy individuals was approved by the Colorado State University Institutional Review Board. All procedures were performed in compliance with the National Institutes of Health’s “Guiding Principles for Ethical Research”. ADSCs at passage 4 were cultured in growth media composed of α-MEM Media with 10% (*v*/*v*) Fetal Bovine Serum and 1% (*v*/*v*) penicillin/streptomycin at 37 °C and under 5% CO_2_ atmosphere [45]. Before cell seeding, the discs were sterilized by incubating them with 70% ethanol for 30 min, followed by 3 rinses with PBS. The cells were then seeded on the sterilized discs at a final concentration of 1.0 × 10^4^ cells/mL.

#### 2.3.2. Cell Toxicity

The cytotoxicity of the alloys was verified using lactate dehydrogenase (LDH) assay (Pierce). After 1 day of cell culture, 100 µL of disc-exposed growth media was added to 100 µL of LDH reaction solution in a 96-well plate. The well plate was then incubated at 37 °C for 30 min. After the incubation, the absorbance of the solutions was measured at 490 nm using a microplate reader (FLUO-Star Omega, BMG Labtech, Ortenberg, Germany). The maximum LDH release was obtained by adding 10% (*v*/*v*) Triton X-100 to wells without any surface to promote cell lysis [46]. Spontaneous LDH release was obtained in the wells in which the cells were not exposed to the discs. The cytotoxicity was calculated following the manufacturer’s guidelines.

#### 2.3.3. Cell Viability

The cell viability of the alloys was evaluated after 4 and 7 days of ADSC culture using CellTiter-Blue assay (Promega, Madison, WI, USA). The discs with adhered cells were incubated at 37 °C in fresh culture media with 10% (*v*/*v*) CellTiter-Blue reagent [45]. After 7 h, the absorbance of the solutions was measured at 570 and 600 nm using a microplate reader (FLUO-Star Omega, BMG Labtech). The CellTiter-Blue reduction percentage was calculated as described by the manufacturer’s protocol, and the cell viability results were normalized using CP Ti outcomes as the positive control (100%).

## 3. Results and Discussion

The composition of the Ti-Mo-Nb-Zr alloy produced was verified by XRF (Table 1). The mass concentration of the elements is next to the projected levels—Ti_10_Mo_8_Nb_6_Zr. Among the contaminants, the concentration of hafnium stands out. Probably the entry of hafnium in the alloy composition occurred through zirconium. The two elements are chemically similar, and therefore, complete separation is a difficult process [47].

The structure present in the alloy at the end of the processing is fundamental for the control of the properties. The alloying elements were used in this study to stabilize the beta-phase after quenching, seeking Young’s modulus lower than that of commercial alloys, such as Ti_6_Al_4_V and CP Ti, among other properties. XRD was used to monitor the evolution of the structures present in the Ti_10_Mo_8_Nb_6_Zr alloy during processing (Figure 1).

The evaluation of the diffractogram revealed the presence of the beta-phase (reference ICSD 76165) in all steps of the processing route. In addition to the beta-phase, the alpha-phase (reference ICSD 197501) was found in the as-cast sample and after homogenization heat treatment, and the alpha”-phase (reference ICSD 105248), after forging. After quenching, only peaks of the beta-phase were observed, which characterizes a beta-type titanium alloy. At the end of the processing, the peak located at 2θ equal to 38.8° became more intense than that observed in the previous steps. The significant increase in the intensity of this peak indicates a preferential orientation to the plane (110) in the BCC structure. This preferential orientation may be related to the forging carried out in the previous step because the XRD carried out after the other processing steps did not indicate such a significant increase in the intensity of this peak.

The lattice parameters and the concentration of the phases during the processing route were evaluated by refining the XRD data using the Rietveld method (Figure 2). Considering the results found in the literature [41,48,49,50,51], the quality parameters of the Rietveld refinement obtained indicate an excellent fit between the observed and calculated data (Table 2).

The Rietveld refinement indicated the beta-phase as the major phase in all steps of the processing route (Table 3). The Ti_10_Mo_8_Nb_6_Zr alloy was designed to stabilize the beta-phase at the end of processing after quenching. However, the results reveal a high beta-phase content even without the rapid cooling characteristic of quenching, which prevents the transformation of beta/alpha. This feature is mainly because of the use of molybdenum and niobium as alloying elements. Zirconium has a smaller and more dependent influence on the beta-phase stabilization of the other alloying elements. Molybdenum is known to be a strong beta-phase stabilizer, superior to other beta-stabilizers, such as niobium. By combining these three alloying elements, there is a significant reduction in the beta-transus temperature (discussed later), which keeps the BCC structure temperatures near room temperature. In addition, niobium decreases the rate of transformation of phases, i.e., the cooling rate required to maintain the beta-structure becomes smaller. For example, the required cooling rate is reduced from 100 °C/s to 0.3 °C/s when comparing Ti15Nb alloy (% at.) with CP Ti [48]. These modifications caused by the alloying elements justify the presence of high levels of beta phase since the beginning of the processing of the Ti_10_Mo_8_Nb_6_Zr alloy.

The as-cast sample showed a concentration of about 20% alpha-phase and 80% beta-phase. After the homogenization heat treatment, the alpha-phase and beta-phase remained present, but there was an increase in the content of the beta-phase to 99.5%. The heat treatment allowed a better distribution of the alloy elements by the ingot, resulting in a more homogeneous structure. The alpha-phase is mainly present on the face of the ingot in contact with the Al_3_O_2_ crucible used during heat treatment. In this region, the cooling rate is lower, which enables the presence of an HCP structure. There were also changes in the lattice parameters of each of the phases due to the processing. There was an increase in the volume of the unit cell of the alpha-phase and a decrease in the volume of the unit cell of the beta-phase. Cooling in the furnace after melting is faster than during homogenization heat treatment. This rapid loss of energy results in a larger BCC cell. On the other hand, the alpha-phase, which is larger than the beta-phase, can expand due to the contraction of the BCC structure during the homogenization treatment [49], resulting in an increase in the alpha-phase lattice parameters and a decrease in the beta-phase.

The next step of the processing route is to perform quenching in water. Rapid cooling provides complete stabilization of the beta-phase and increases unit cell volume. The forging of the ingots causes increased stress and distortions in the structure, resulting in the appearance of the alpha-phase and martensitic phases, in the content of approximately 13%. This new structure presented a large volume of the unit cell, which, together with the deformations generated by the forging, caused a small contraction of the unit cell of the beta-phase. To return to the monophasic condition with a BCC structure, new quenching was carried out. Therefore, a single structure (BCC) was obtained at the end of the processing, characterizing the Ti_10_Mo_8_Nb_6_Zr alloy as a beta-type titanium alloy.

As previously mentioned, the alloying elements alter the beta-transus transformation temperature, which consequently influences the phases present in the alloy. The XRD revealed that the beta-phase was obtained as the majority phase, even under conditions of slow cooling. It can be assumed that there was a significant decrease in the beta-transus temperature and the transformation rate. Thus, the DSC was performed to understand the effect of the alloying elements on the beta-transus temperature of the Ti_10_Mo_8_Nb_6_Zr alloy (Figure 3a). Endothermic and exothermic peaks and baseline alterations were observed in the alloy analysis; therefore, first and second-order transitions occur. These peaks are related to phase transformations; according to Bönisch et al. [52], endothermic-type transformations give rise to the beta-phase, while its decomposition generates exothermic peaks. However, continuous changes in the crystal lattice involving similar structures can result in second-order transitions recorded in the DSC curves as a baseline change. The transformation between martensitic phases, alpha”/alpha’, in titanium alloys can occur without generating the characteristic discontinuities of first-order transitions. Less commonly, the decomposition of the beta-phase to alpha”-phase can also occur as a second-order transition [9]; there is also the possibility of not being detected by the DSC, as it involves a small enthalpy and is a long transformation [53].

Figure 3 shows the two heating-cooling cycles performed for a sample of the Ti_10_Mo_8_Nb_6_Zr alloy. During the first heating, an exothermic peak was observed at 679 °C (Table 4). This peak proved to be more energetic than the other events observed in this analysis. The sample starts with a single-phase condition with a BCC-type structure. Therefore, this peak does not indicate an alpha/beta transformation since an endothermic event should be verified. There is probably a relief from internal stresses generated from alloy processing and sample preparation. During cooling, a baseline shift at 593 °C and an exothermic peak at 317 °C were observed, indicating the decomposition of the beta-phase into martensitic phases and possibly the precipitation of a small amount of alfa-phase. During heating, an endothermic peak is formed, characteristic of the formation of the beta-phase, so there is a record of the beta-transus temperature at 514 °C. In the second cooling, at 448 °C, second-order transitions are noted, generating martensitic phases.

Molybdenum and niobium are known beta-stabilizers, which have a very strong effect on the beta-transus temperature, which can be observed in the thermal analysis of the Ti_10_Mo_8_Nb alloy [43]. The decrease in beta-transus temperature when adding zirconium, although less expressive, is in line with the behavior reported in the literature, which observes an influence of zirconium in the stabilization of the beta-phase when other beta-stabilizer elements are present in the composition [17,18]. Another aspect observed is about the types of transformations that occurred. All decomposition of the beta-phase in the Ti_10_Mo_8_Nb alloy, observed in a previous study [43], are of first-order transitions; therefore, zirconium should also influence the transformation mechanism of the structure.

The microstructure showed a typical beta-phase morphology at all steps of the processing route evaluated (Figure 4). The as-cast sample showed regions of the formation of a dendritic microstructure on the face of the ingot in contact with the copper crucible used during the melting (Figure 4a). The combination of a high cooling rate and a single-phase region leads to the formation of this type of microstructure. In this region of the ingot, the cooling rate is higher as the crucible undergoes forced cooling by the flow of water, producing a dendritic microstructure. Equiaxial grain characteristics of the beta-phase were observed in the rest of the ingot. After the homogenization heat treatment, there was a growth of the grains due to slow cooling (Figure 4b). The better distribution of the elements also provided a greater homogeneity of the microstructure. After quenching, a microstructure similar to that found in the previous condition, characteristic of the beta-phase, was observed (Figure 4c). However, there was a decrease in the average grain size due to rapid cooling. In the micrograph after forging, deformation lines were noted (Figure 4d). At the end of the processing, a microstructure similar to that found after the first quenching was found. The average grain size at the end of the processing route was 228.7 μm, similar to that obtained in a study with the Ti_10_Mo_8_Nb alloy, 220.7 μm [43]. Therefore, the addition of zirconium had little influence on the recrystallization of grains.

The evaluation of mechanical properties was evaluated by measuring the microhardness and Young’s modulus of the Ti_10_Mo_8_Nb_6_Zr alloy. Secondary phases influenced the hardness of the Ti_10_Mo_8_Nb_6_Zr alloy, as there was a significant difference between the average values measured at each stage of the processing route (Figure 5). Only the measurements after the first quenching and at the end of the processing did not show a statistical difference, as they are similar conditions. The hardness of the samples that presented exclusively beta-phase was less than that of the other conditions, which is a condition reported in the literature [27]. The microhardness of the Ti_10_Mo_8_Nb_6_Zr alloy was superior to that of the CP Ti (185 HV) and similar to that of the Ti_6_Al_4_V alloy [54]. The increase in hardness is due to the alloying elements, mainly zirconium [21]. Young’s modulus was obtained by the impulse excitation technique. The measurements were carried out in the transversal direction (flexural method) and the longitudinal direction (longitudinal method). There was no significant difference between the measurements performed, indicating that the processing route resulted in the homogeneity of the property. The values obtained were: 83.5 GPa in the transversal direction and 83.8 in the longitudinal direction. Therefore, Young’s modulus is lower than that of CP Ti (115 GPa) and Ti_6_Al_4_V alloy (110 GPa) [1,4], a positive result for bone tissue applications. Zirconium influenced the drop in Young’s modulus of the studied alloy, as the preceding ternary alloy, Ti_10_Mo_8_Nb, when produced by the same processing route, showed a superior Young’s modulus: 90 GPa [43].

For possible use in biomedical devices, the alloy must have acceptable levels of toxicity and, in certain applications, stimulate cell proliferation. Thus, two in vitro tests were performed to observe the level of cytotoxicity and cell proliferation of the Ti_10_Mo_8_Nb_6_Zr alloy compared to CP Ti and the Ti_10_Mo_8_Nb alloy. The level of cytotoxicity was evaluated by the presence of the enzyme Lactate dehydrogenase (LDH) on the surface of the samples. The enzyme is present in living cells and is released into the environment when the cell is damaged [45]. No significant difference was detected between the average percentage of LDH for the materials evaluated (Figure 6). Therefore, the Ti_10_Mo_8_Nb_6_Zr alloy has a toxicity level similar to that of CP Ti, a material that has biocompatibility levels considered excellent [4]. According to ISO 10993-5, a biomaterial is considered cytotoxic if the cytotoxicity values are greater than 10%. Therefore, the alloys can be considered non-cytotoxic. Cell viability was verified through tests performed with the Alamar Blue reagent. This reagent is permeable to cells, and it indicates mitochondrial activity. Thus, the reduction of the reagent indicates cell growth [45]. The evaluation was performed after four and seven days of incubation with the cells. The results obtained are favorable for the Ti_10_Mo_8_Nb_6_Zr alloy, which showed higher cell viability values than the CP Ti and the Ti_10_Mo_8_Nb alloy. After seven days, there was a significant increase in cell viability, which indicates the greater mitochondrial activity of cells in contact with the Ti_10_Mo_8_Nb_6_Zr alloy (Figure 7).

These results indicate a potential for application in medical devices, with results superior to CP Ti. According to the literature, niobium has excellent biocompatibility [10], but in this study, zirconium had a great influence on the results, even though the alloying element was at the lowest concentration. However, the addition of 6%wt of zirconium to the alloy composition was sufficient to generate superior results, highlighting its potential for biomedical applications.

## 4. Conclusions

The Ti_10_Mo_8_Nb_6_Zr alloy is a beta-type titanium alloy. Throughout the processing was observed the presence of the beta-phase was observed as the majority phase. The alpha-phase is present after melting and after the homogenization heat treatment. Cold forging resulted in the formation of the alpha”-phase. At the end of the processing route, the Ti_10_Mo_8_Nb_6_Zr alloy presented only the beta-phase. Zirconium influenced the decrease in beta-transus temperature and Young’s modulus and increased biocompatibility. The Ti_10_Mo_8_Nb_6_Zr alloy showed a beta-transus temperature of 407 °C. The microstructure is characteristic of the beta-phase, with equiaxial grains and an average size of 229 μm. There was an increase in the hardness of CP Ti and similarity to that found in Ti_6_Al_4_V. Young’s modulus was 83 GPa, which is lower than CP Ti and Ti_6_Al_4_V alloys. There was no significant variation as a measure of Young’s modulus performed in the transverse and longitudinal direction of the ingot, showing homogeneity of the property. The Ti_10_Mo_8_Nb_6_Zr alloy showed non-cytotoxicity, and cell viability was higher than that obtained for CP Ti. Thus, the Ti_10_Mo_8_Nb_6_Zr alloy showed properties that give it potential for applications in biomedical devices.

## Figures and Tables

**Figure 1 materials-15-08636-f001:**
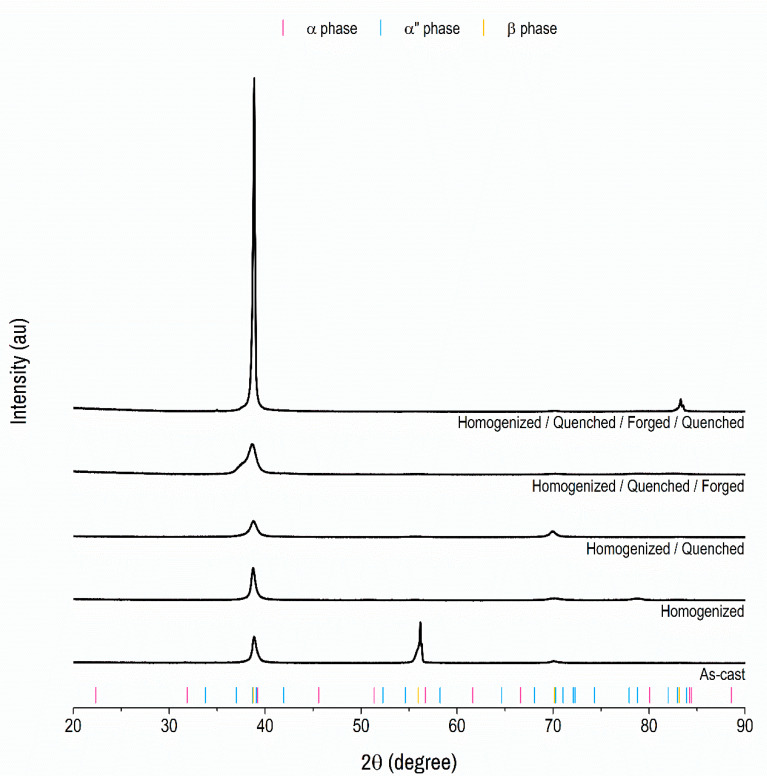
**X-ray diffraction pattern of the Ti_10_Mo_8_Nb_6_Zr alloy.** The black lines represent the X-ray diffraction pattern of the Ti_10_Mo_8_N_6_Zr alloy at each processing stage, from the melting to the final stage. The pink bars represent the X-ray diffraction pattern of the alpha-phase, according to reference ICSD 197501. The blue bars represent the X-ray diffraction pattern of the alpha” phase, according to reference ICSD 105248. The yellow bars represent the X-ray diffraction pattern of the beta phase, according to reference ICSD 76165.

**Figure 2 materials-15-08636-f002:**
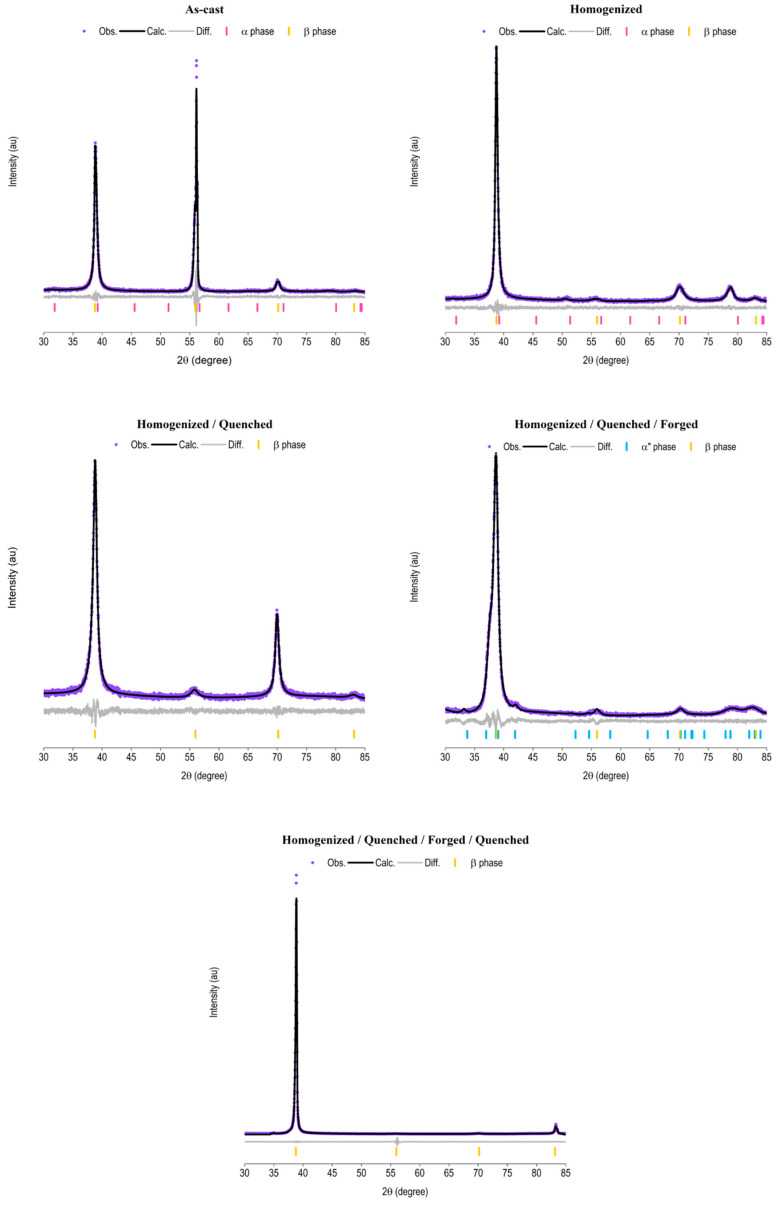
**Rietveld refinement of the X-ray diffraction pattern of the Ti_10_Mo_8_Nb_6_Zr alloy.** The analysis was carried out after each processing step of the alloy: (**a**) as cast, (**b**) after homogenization heat treatment, (**c**) after quenching, (**d**) after cold forging, and (**e**) at the end of processing. The purple circles represent the observed XRD data. The black lines represent data calculated by Rietveld refinement. The gray lines represent the difference between the observed data and data calculated by Rietveld refinement. The pink bars represent the X-ray diffraction pattern of the alpha-phase, according to reference ICSD 197501. The blue bars represent the X-ray diffraction pattern of the alpha” phase, according to reference ICSD 105248. The yellow bars represent the X-ray diffraction pattern of the beta phase, according to reference ICSD 76165.

**Figure 3 materials-15-08636-f003:**
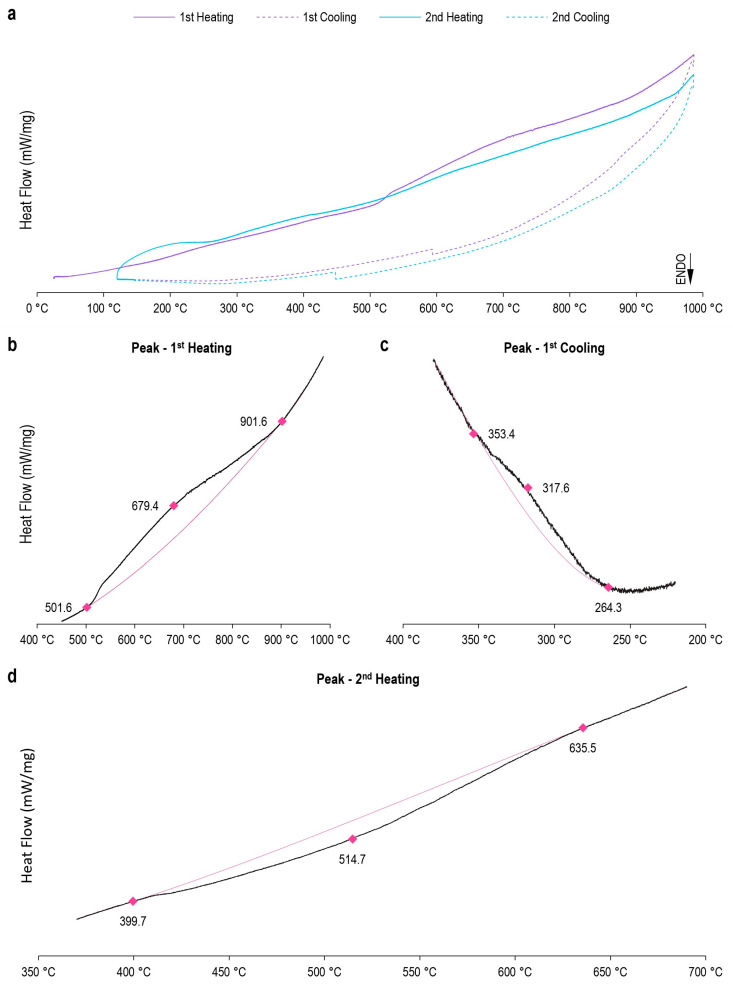
**DSC of the Ti_10_Mo_8_Nb_6_Zr alloy.** (**a**) Plotting the two heating-cooling cycles performed on a sample of Ti_10_Mo_8_Nb alloy in the final processing condition. The first cycle is indicated by purple. The second cycle is indicated by blue. The solid lines represent the heating. The dashed lines represent the cooling. Magnification of the area in which thermal events occur during (**b**) the first heating, (**c**) the first cooling, and (**d**) the second heating. The black lines represent the observed data. The pink lines represent the baseline. The initial, peak, and end temperatures of the thermal events are indicated.

**Figure 4 materials-15-08636-f004:**
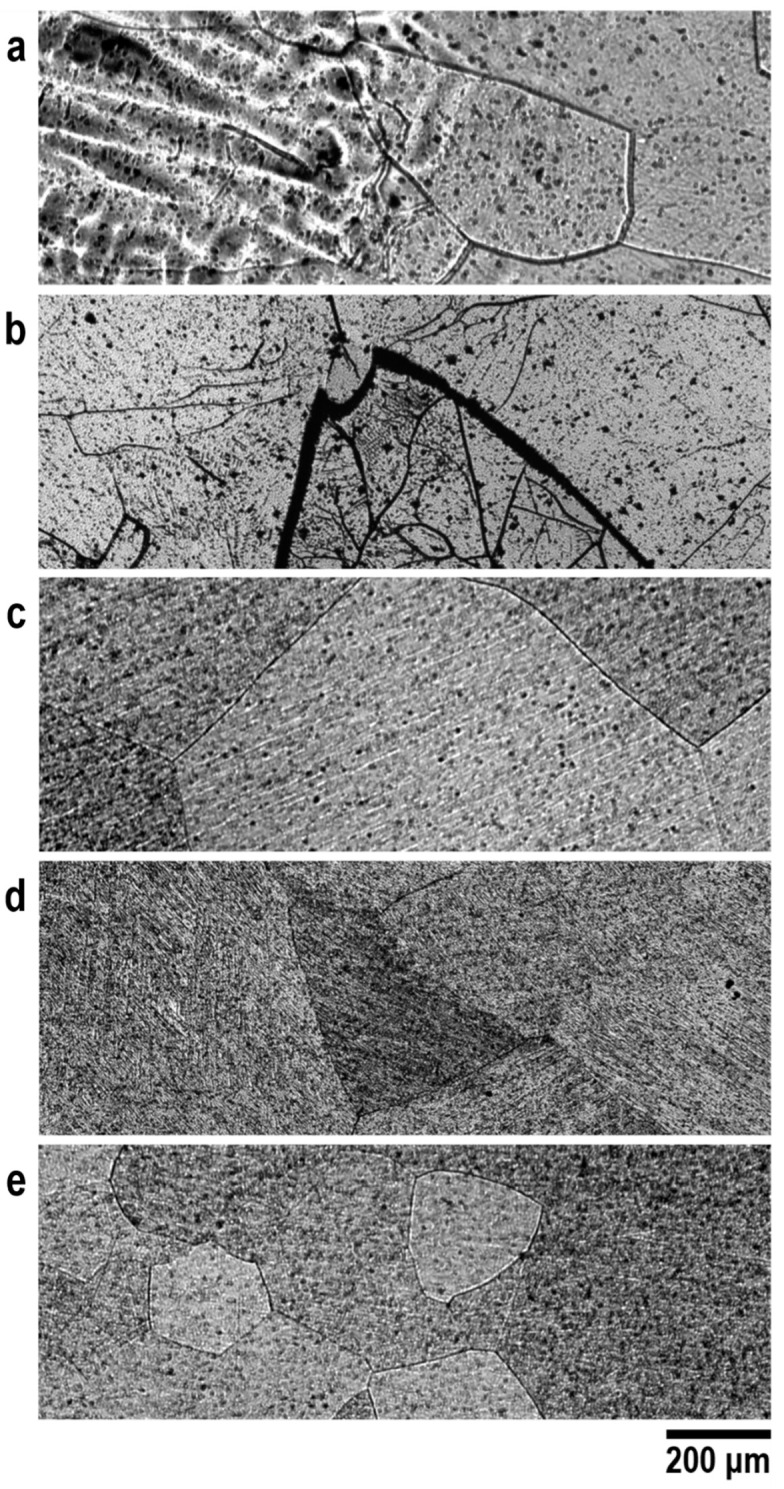
**Optical micrographs of the Ti_10_Mo_8_Nb_6_Zr alloy**. Imagens of the microstructure of the Ti_10_Mo_8_Nb_6_Zr alloys at different stages of processing: (**a**) as cast, (**b**) after homogenization heat treatment, (**c**) after quenching, (**d**) after cold forging, and (**e**) at the end of processing.

**Figure 5 materials-15-08636-f005:**
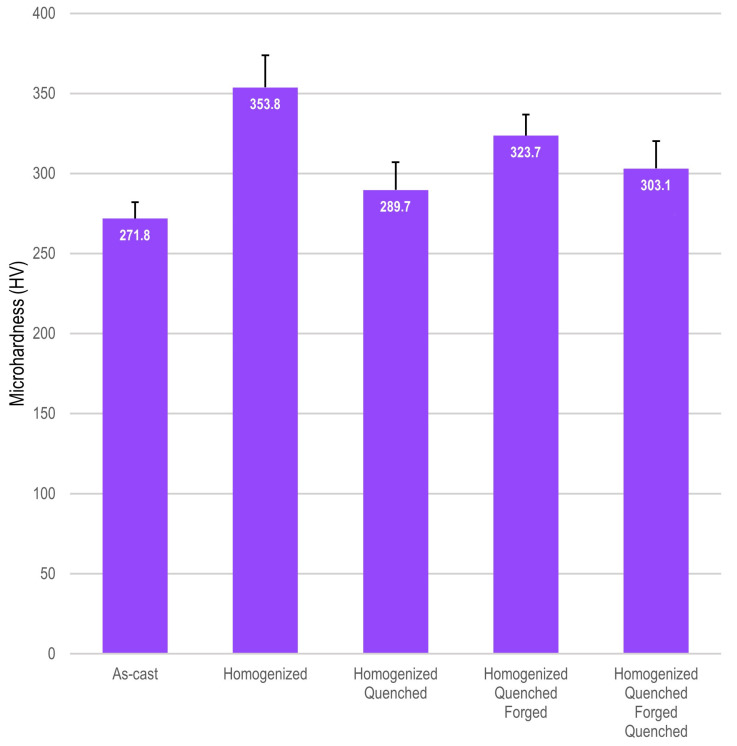
**Microhardness of the Ti_10_Mo_8_Nb_6_Zr**. The columns indicate the average Vickers microhardness evaluated for each stage of processing, and the bars indicate standard deviation.

**Figure 6 materials-15-08636-f006:**
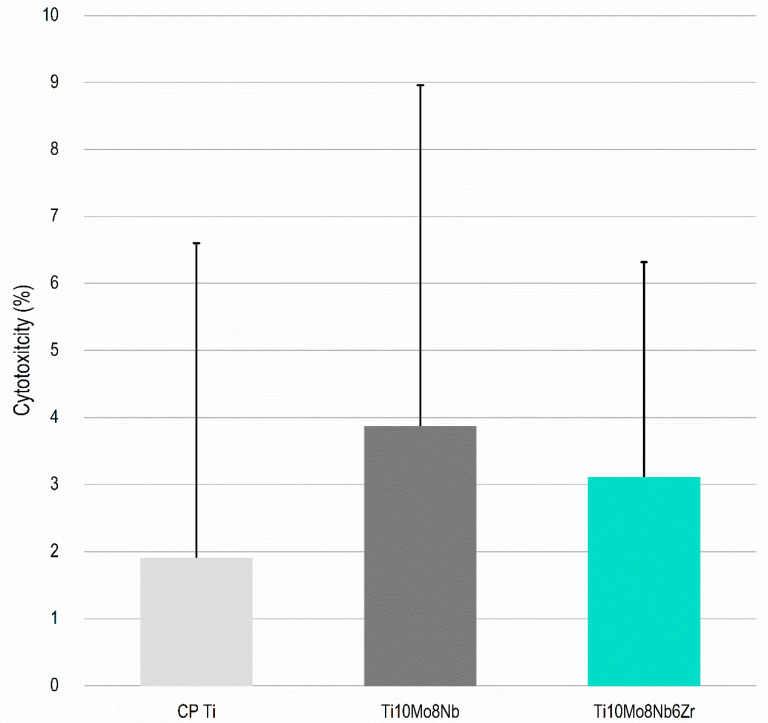
**Cytotoxicity evaluation of the Ti_10_Mo_8_Nb_6_Zr alloy.** LDH analysis on surfaces of the CP Ti, Ti_10_Mo_8_Nb, and Ti_10_Mo_8_Nb_6_Zr alloys after one day of cell culture. The columns indicate the average values, and the bars indicate the standard deviation.

**Figure 7 materials-15-08636-f007:**
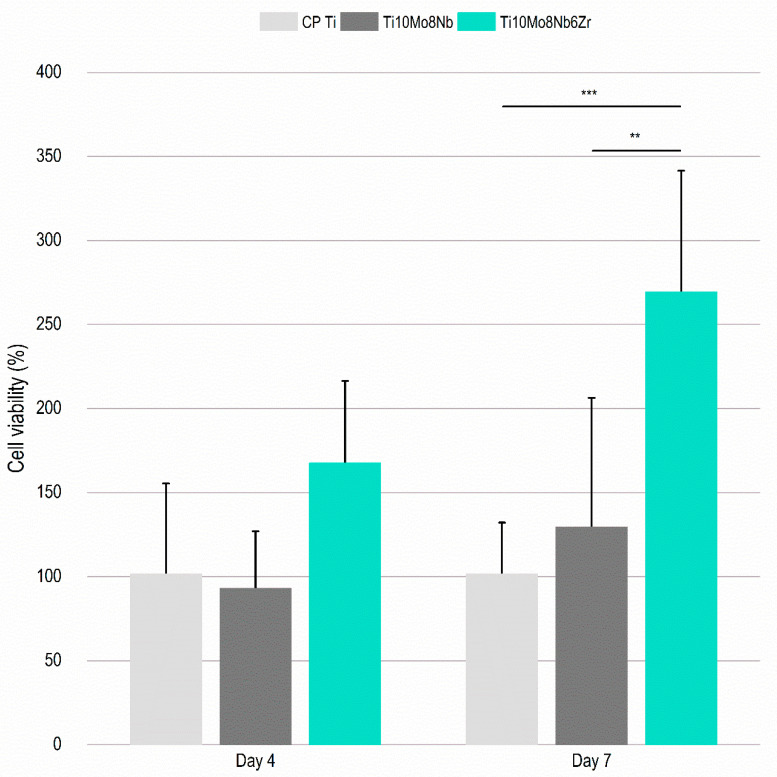
**Cell growth on the surface of the Ti_10_Mo_8_Nb_6_Zr alloy.** Cell viability after four and seven days of ADSC culture about the growth of cells on the surface of the CP TI about the cells on the surface of the Ti_10_Mo_8_Nb and Ti_10_Mo_8_Nb_6_Zr alloys. (**) *p* < 0.01, (***) *p* < 0.001 using the Mann–Whitney non-parametric test for independent samples, SPSS 13.0 software. The columns indicate the average values, and the bars indicate the standard deviation.

**Table 1 materials-15-08636-t001:** The chemical composition of Ti-Mo-Nb-Zr produced alloy.

Element	Ti	Mo	Nb	Zr	Hf	Al	Na	Fe	Si
**Mass concentration**	Balance	11.17%	8.80%	6.48%	0.66%	0.30%	0.13%	0.06%	0.04%

**Table 2 materials-15-08636-t002:** Quality parameters of the Rietveld refinement.

Ti_10_Mo_8_Nb_6_Zr	R_p_	R_wp_	R_exp_	GOF	R-Bragg	
As-cast	6.26	8.09	6.00	1.35	alpha-phase	0.649
					beta-phase	0.491
Homogenized	4.95	6.64	6.34	1.05	alpha-phase	0.500
					beta-phase	0.225
Homogenized/Quenched	5.40	6.95	6.39	1.09	beta-phase	0.343
Homogenized/Quenched/Forged	5.60	7.71	5.77	1.30	alpha“-phase	1.361
					beta-phase	0.531
Homogenized/Quenched/Forged/Quenched	6.11	7.98	4.34	1.84	beta-phase	2.137

**Table 3 materials-15-08636-t003:** The concentrations of the phases and lattice parameters were obtained by Rietveld refinement.

Ti_10_Mo_8_Nb_6_Zr	Alpha-Phase			Alpha″-Phase		Beta-Phase	
	Conc. (%)	a = b (Å); c (Å)	c/a	Conc. (%)	a (Å); b (Å); c (Å)	Conc. (%)	a = b = c (Å)
As-cast	19.66	4.64; 2.84	1.63	-	-	80.34	3.27
Homogenized	0.55	4.65; 2.86	1.62	-	-	99.45	3.26
Homogenized/Quenched	-	-	-	-	-	100.00	3.28
Homogenized/Quenched/Forged	-	-	-	12.84	3.25; 4.81; 4.70	87.16	3.28
Homogenized/Quenched/Forged/Quenched	-	-	-	-	-	100.00	3.30

**Table 4 materials-15-08636-t004:** Data were obtained from the DSC of the Ti_10_Mo_8_Nb_6_Zr alloy.

Stages		T_initial_ (°C)	T_peak_ (°C)	T_final_ (°C)	Entalpy (J/g)
1st Heating	1st order phase transition	501.6	679.4	901.6	−30.0
1st Cooling	2nd order phase transition	593.0	-	-	-
	1st order phase transition	353.4	317.6	264.0	−0.2
2nd Heating	1st order phase transition	399.7	514.7	635.5	4.3
2nd Cooling	2nd order phase transition	448.28	-	-	-

## Data Availability

Not applicable.

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
