# Peer review of "Processing and Characterization of a New Quaternary Alloy Ti10Mo8Nb6Zr for Potential Biomedical Applications"

_materials, 2022, doi:10.3390/ma15238636_

Round 1

Reviewer 1 Report

The manuscript contains publishable information and add to the field. Just two minor recommendations are given:

-In page 2, line 93 of the introduction, the sentence starts by "Therefore, our aim is...." is incomplete.

- Some results are taken from the thesis of "https://repositorio.unesp.br/handle/11449/181754 ", please add to the refrences

-

Author Response

RESPONSE TO REVIEWER COMMENT

Manuscript ID: materials-2019430

Article Title: Processing and characterization of a new quaternary alloy Ti10Mo8Nb6Zr for potential biomedical applications

The authors are grateful for the reviewer's contributions. Changes or justifications to the reported points are set out below.

The manuscript contains publishable information and add to the field. Just two minor recommendations are given:

-In page 2, line 93 of the introduction, the sentence starts by "Therefore, our aim is...." is incomplete.

The text has been changed to better understanding.

Before: “Therefore, our objective in the present study was to evaluate the influence of the addition of zirconium on the Ti10Mo8Nb ternary system previously evaluated by our group”

After: “Therefore, our objective in the present study was to evaluate the influence of zirconium addition on the properties of the ternary Ti10Mo8Nb system previously evaluated by our group”

- Some results are taken from the thesis of "https://repositorio.unesp.br/handle/11449/181754 ", please add to the references

The references cited by the reviewer were replaced by a new paper, published after the manuscript was written.

Before:

[43] J.P.A. Carobolante, Surface modification of the esperimental alloy Ti10Mo8Nb employing anodic oxidation: in vitro studies, Thesis, São Paulo State University (Unesp), 2017. http://hdl.handle.net/11449/150122.

[55] A. Pereira Júnior, Microstructural and mechanical characterization of the Ti10Mo8Nb6Zr alloy for biomedical applications, Thesis, São Paulo State University (UNESP), 2019. http://hdl.handle.net/11449/181754.

After:

[43] P. Capellato, F.B. Vilela, A.H.P. Fontenele, K.B. da Silva, J. P.A. Carobolante, E.G.M. Bejarano, M. de Lourdes Noronha Motta Melo, A.P.R.A. Claro, D. Sanchs, Evaluation of microstructure and mechanical properties of a Ti10Mo8Nb alloy for biomedical applications. Metals. 12 (2022). 1065. https://foi.org/10.3390/met12071065.

Reviewer 2 Report

An important application of metallic biomaterials is their use in medical devices to aid musculoskeletal disabilities, which are the main cause of life with a disability. Ti10Mo8Nb6Zr beta-type titanium alloy appears to be such promising metallic biomaterial. The collection of results obtained by X-ray fluorescence spectroscopy, differential scanning calorimetry, optical microscopy, microhardness measurements, and pulse excitation techniques constitute a convincing material diagnosis. The microscopic observations are rather optimistic. For example, based on the analysis of optical micrographs, the microstructure of the Ti10Mo8Nb6Zr alloy after cold forging showed a typical beta phase morphology, and an equilibrium distribution of deformation lines was found. Similarly, in spite of controversy regarding the biocompatibility of molybdenum, the alloy of interest discloses an acceptable level of toxicity (LDH in vitro test) and stimulates cell proliferation. Cell viability was verified through tests performed with the Alamar Blue reagent. Finally, The Ti10Mo8Nb6Zr alloy showed cell viability higher than that obtained for CP Ti. However, X-ray diffraction results and conclusions arouse controversy.

1)     Comments on Figures 1 and 2: Authors discuss here the phases coexisting together with the basic Beta one for which the reference card code should be written as 76165.

2)     At the main text directly below the figure it can find the information that ... In addition to the beta-phase, the alpha-phase (reference ICSD 197501) was found in the as-cast sample and after homogenization heat treatment, ...

3)     I don’t really see any other contribution than the alpha” phase. Please explain this. Refinements presented in Figures 2a-e are unreadable. All should be corrected. It is recommended to limit the Intensity scale by a factor of at least 2.

4)     The precision of the unit cells parameters presented in Table 3 is incredibly high. Please check and comment.

5)     Page 9, line 250: …The as-cast sample showed a concentration of about 20% alpha-phase and 80% beta- phase. What kind of concentration do Authors mean?

Author Response

RESPONSE TO REVIEWER COMMENT

Manuscript ID: materials-2019430

Article Title: Processing and characterization of a new quaternary alloy Ti10Mo8Nb6Zr for potential biomedical applications

The authors are grateful for the reviewer's contributions. Changes or justifications to the reported points are set out below.

An important application of metallic biomaterials is their use in medical devices to aid musculoskeletal disabilities, which are the main cause of life with a disability. Ti10Mo8Nb6Zr beta-type titanium alloy appears to be such promising metallic biomaterial. The collection of results obtained by X-ray fluorescence spectroscopy, differential scanning calorimetry, optical microscopy, microhardness measurements, and pulse excitation techniques constitute a convincing material diagnosis. The microscopic observations are rather optimistic. For example, based on the analysis of optical micrographs, the microstructure of the Ti10Mo8Nb6Zr alloy after cold forging showed a typical beta phase morphology, and an equilibrium distribution of deformation lines was found. Similarly, in spite of controversy regarding the biocompatibility of molybdenum, the alloy of interest discloses an acceptable level of toxicity (LDH in vitro test) and stimulates cell proliferation. Cell viability was verified through tests performed with the Alamar Blue reagent. Finally, The Ti10Mo8Nb6Zr alloy showed cell viability higher than that obtained for CP Ti. However, X-ray diffraction results and conclusions arouse controversy.

1)     Comments on Figures 1 and 2: Authors discuss here the phases coexisting together with the basic Beta one for which the reference card code should be written as 76165.

Reference to card code changed from 076165 to 76165.

2)     At the main text directly below the figure it can find the information that ... In addition to the beta-phase, the alpha-phase (reference ICSD 197501) was found in the as-cast sample and after homogenization heat treatment, ...

3)     I don’t really see any other contribution than the alpha” phase. Please explain this. Refinements presented in Figures 2a-e are unreadable. All should be corrected. It is recommended to limit the Intensity scale by a factor of at least 2.

The alpha” and beta phases present, in certain regions of their X-ray diffraction pattern, very close values of 2θ. In addition, the peaks obtained by the X-ray diffractometer in this type of material are not extremely defined, presenting relatively wide bases. In this way, there may be a contribution, even if little, from more than one phase in its formation.

This interpretation was corroborated by the evaluation of the results obtained by the Rietveld refinement method, the results were always better when the refinement was carried out with the pattern of alpha” and beta phases.

Figures 2a-e have been changed for better readability: strokes and fonts have been enlarged.

4)     The precision of the unit cells parameters presented in Table 3 is incredibly high. Please check and comment.

The results presented were those obtained by Rietveld refinement in the TOPAS software, Bruker. In order to establish a standard among all presented parameters, the data in the table has been changed to two decimal places.

5)     Page 9, line 250: …The as-cast sample showed a concentration of about 20% alpha-phase and 80% beta- phase. What kind of concentration do Authors mean?

The content of the phases present in the sample was obtained from the refinement by the Rietveld method of the data obtained by diffractometry, using the TOPAS software. Thus, as the technique evaluates a certain volume of the sample, the concentrations of the phases present are related to the volume of the analyzed region. That is, when the concentration of the phases is mentioned in the text, the values refer to the estimation of the volumetric concentration of these phases.

Reviewer 3 Report

The authors investigated a new titanium alloy for implant with Young’s modulus 83GPa, decreased much than pure Ti and other alloys. The processing procedure, heat treatment, mechanical test and biocompatiable properties tests were conducted. The results showed the superiority of this alloy.

I have two suggestions to the authors:

1. The SEM photos of cell growth should be given to prove the alloy potential on cell viability.

2. The bone's Young’s modulus is no more than 20GPa, which is still much lower than the new alloy (83GPa), so how to avoid the stress shield when implant of new alloy is implanted?

Author Response

RESPONSE TO REVIEWER COMMENT

Manuscript ID: materials-2019430

Article Title: Processing and characterization of a new quaternary alloy Ti10Mo8Nb6Zr for potential biomedical applications

The authors are grateful for the reviewer's contributions. Changes or justifications to the reported points are set out below.

The authors investigated a new titanium alloy for implant with Young’s modulus 83GPa, decreased much than pure Ti and other alloys. The processing procedure, heat treatment, mechanical test and biocompatible properties tests were conducted. The results showed the superiority of this alloy.

I have two suggestions to the authors:

  1. The SEM photos of cell growth should be given to prove the alloy potential on cell viability.

We agree with the reviewer’s comment and conduct these experiments in future since it is beyond the scope of this manuscript.

  1. The bone's Young’s modulus is no more than 20GPa, which is still much lower than the new alloy (83GPa), so how to avoid the stress shield when implant of new alloy is implanted?

The Ti10Mo8Nb6Zr alloy showed a significant decrease in Young's modulus when compared to commercial alloys - CP Ti and Ti6Al4V - applied in the biomedical field. The value is still far from the range found for bone tissue, but these small advances should contribute to the reduction of bone atrophy caused by the stress shielding effect. In addition, other surface treatments can be applied to this new alloy, such as anodization [https://doi.org/10.1016/j.surfcoat.2020.125467; https://doi.org/10.1680/jsuin.16.00011; https://doi.org/10.1590/1980-5373-MR-2017-0508], effectively for better adhesion and development of bone tissue next to the improved material.

Round 2

Reviewer 2 Report

Controversial issues have been thoroughly explained by the Authors.

Necessary adjustments have also been included.

The work in its current form does not raise any objections.